# Trimethylamine N-Oxide Exacerbates Neuroinflammation and Motor Dysfunction in an Acute MPTP Mice Model of Parkinson’s Disease

**DOI:** 10.3390/brainsci13050790

**Published:** 2023-05-12

**Authors:** Wei Quan, Chen-Meng Qiao, Gu-Yu Niu, Jian Wu, Li-Ping Zhao, Chun Cui, Wei-Jiang Zhao, Yan-Qin Shen

**Affiliations:** Department of Neurodegeneration and Injury, Wuxi School of Medicine, Jiangnan University, Wuxi 214122, China; 6202805016@stu.jiangnan.edu.cn (W.Q.);

**Keywords:** Parkinson’s disease, trimethylamine N-oxide, gut microbiota metabolites, neuroinflammation, microglia, astrocytes

## Abstract

Observational studies have shown abnormal changes in trimethylamine N-oxide (TMAO) levels in the peripheral circulatory system of Parkinson’s disease (PD) patients. TMAO is a gut microbiota metabolite that can cross the blood–brain barrier and is strongly related to neuroinflammation. Neuroinflammation is one of the pathological drivers of PD. Herein, we investigated the effect of TMAO on 1-methyl-4-phenyl-1,2,3,6-tetrahydropyridine (MPTP)-induced PD model mice. TMAO pretreatment was given by adding 1.5% (*w*/*v*) TMAO to the drinking water of the mice for 21 days; then, the mice were administered MPTP (20 mg/kg, i.p.) four times a day to construct an acute PD model. Their serum TMAO concentrations, motor function, dopaminergic network integrity, and neuroinflammation were then assayed. The results showed that TMAO partly aggravated the motor dysfunction of the PD mice. Although TMAO had no effect on the dopaminergic neurons, TH protein content, and striatal DA level in the PD mice, it significantly reduced the striatal 5-HT levels and aggravated the metabolism of DA and 5-HT. Meanwhile, TMAO significantly activated glial cells in the striatum and the hippocampi of the PD mice and promoted the release of inflammatory cytokines in the hippocampus. In summary, higher-circulating TMAO had adverse effects on the motor capacity, striatum neurotransmitters, and striatal and hippocampal neuroinflammation in PD mice.

## 1. Introduction

Parkinson’s disease (PD) is the second most common neurodegenerative disease, and its pathological mechanism remains unclear [1]. However, there is increasing evidence that the gut microbiota and its metabolites can influence the central nervous system via the microbiota–gut–brain axis [2,3]. Previous studies in our laboratory have found that transplanting the gut microbiota of normal mice into PD model mice can significantly improve the behavioral disorders and brain pathology of PD mice [4]. Vancomycin can exert neuroprotective effects on PD mice by regulating the gut microbiota and inhibiting gut and brain inflammation [5]. In addition, in 2021, Hou et al. found that propionic acid (a metabolite of the gut microbiota) could reverse motor deficits and dopaminergic neuron loss in 6-OHDA-induced PD mice [6]. Together, these studies suggest that the gut microbiota and its metabolites play an important role in PD pathology.

Trimethylamine N-oxide (TMAO) is an indirect metabolite of the gut microbiota [7]. Studies have shown that TMAO can cross the blood–brain barrier (BBB) and impair its integrity [8]. Recent clinical studies found that the plasma TMAO levels of PD patients were significantly higher than those of healthy controls, and higher TMAO levels were positively correlated with the severity and progression of PD [9]. However, Chung et al. reported that the plasma TMAO levels of PD patients were significantly lower than those of the control group, and lower levels of TMAO could be used as a biomarker for early PD [10]. In addition, plasma TMAO levels were significantly higher in middle-aged and older adults compared to younger adults, and higher TMAO levels were associated with cognitive decline [11]. In both animal and cell studies, TMAO has been found to impair cognitive performance in young mice, induce neuroinflammation, and increase the release of proinflammatory cytokines in the brain [11,12]. In conclusion, there is a close relationship between TMAO and central nervous system diseases. In particular, clinical studies have shown that abnormal changes in TMAO may have a causal relationship with the occurrence and development of PD. However, the existing studies are limited and have different conclusions.

Based on the above research background, to clarify the effect of TMAO on PD, we used the MPTP-induced acute PD model to assess the effect of higher circulating TMAO on PD-related pathologies, such as neuroinflammation and dopaminergic network integrity. The MPTP-induced acute PD model, which exhibits stable and effective loss of dopaminergic neurons and intense inflammatory responses in the striatum and SN, is one of the classic animal models of PD [13]. We increased peripheral TMAO levels by adding TMAO to the drinking water of the mice and then injected MPTP intraperitoneally to induce acute PD model mice. Finally, the striatal tissue, substantia nigra tissue, and hippocampal DG in the acute PD model mice with or without TMAO treatment were analyzed.

## 2. Materials and Methods

### 2.1. Animal Grouping, PD Model Construction, and TMAO Treatment

The six-week-old male C57BL/6J mice were provided by Vital River Laboratory Animal Technology (Tongxiang, China). All of the mice were acclimatized for 7 days before treatment and were maintained under a specific pathogen-free environment at the Medical Laboratory Animal Center of Jiangnan University. The room was kept at 24 ± 2.0 °C and 55 ± 10% humidity with a standard diurnal cycle. The mice had free access to drinking water and a standard diet. All of the animal procedures were performed in accordance with the guideline of the Jiangnan University Animal Care Committee.

The mice were randomly divided into four groups: (1) the drinking water supplemented with 1.5% TMAO + intraperitoneal injection of normal saline control group (TMAO group); (2) the saline-treated group (the control group); (3) the drinking water supplemented with 1.5% TMAO + intraperitoneal injection of MPTP to construct the model group (TMAO + MPTP group); and (4) the Parkinson’s disease model group (MPTP group). During the whole experimental period, the four groups of mice were fed with the same diet. The tissues were collected on the 28th day. Approximately 1.5% TMAO [14] (M37603, Meryer (Shanghai) Chemical Technology, Shanghai, China) was given in the drinking water of the mice for 28 days, and the water was renewed every two days. On the 21st day of the experiment, the PD model mice were established via intraperitoneal injection of MPTP (M0896, Sigma-Aldrich, St. Louis, MO, USA) (20 mg/kg). The intraperitoneal injections were given four times a day at two-hour intervals. The experimental design diagram is shown in Figure 1A.

### 2.2. Liquid Chromatography–Mass Spectrometry (LC–MS)

For sample preprocessing before LC–MS detection, 150 μL acetonitrile containing the labelled internal standards (Trimethylamine-d9 N-Oxide, T86572, Shanghai Yuanye Bio-Technology, Shanghai, China) was added into 50 μL serum in 1.5 mL Eppendorf tubes to precipitate the protein. Then, the tube was vortexed for 1 min and centrifuged at 13,200 rpm for 15 min at 4 °C. The supernatant was passed through the filter (0.22 μm) and then it was detected by high performance liquid chromatography with electrospray ionization tandem mass spectrometry (Vanquish Q Exactive Plus, Thermo Scientific, Waltham, MA, USA). ACQUITY UPLC^®^BEH C18 column (2.1 mm × 100 mm, 1.7 μm). For details of the detection method, please refer to the previous report [15].

### 2.3. Behavioral Tests

To evaluate the motor function of mice in each group, the mice were firstly subjected to behavioral training for three consecutive days from days 25 to 27, as indicated in the protocol. Then, behavioral tests (including the pole test and traction test) were performed on day 28. The training and testing were conducted at the same time each morning (8:00 a.m.). In the pole test, a thick rod of 1 cm in diameter and 50 cm in length was first placed in the center of the empty cage, and the time from the top of the rod to the bottom of the cage was recorded for each mouse. The average value of three independent tests was taken for analysis. In the traction test, the ability of the mice to grasp a 0.5 cm-diameter, horizontal wire was evaluated. The score criteria were as follows: all limbs can grasp, recorded as 4 points; two forelimbs and one hind limb grip, recorded as 3 points; two forelimbs grasp, recorded as 2 points; only one forelimb can grasp, recorded as 1 point; unable to grasp and dropped the wire, recorded as 0 points. Each test was performed independently three times, and the average score of the three tests was calculated for statistical analysis.

### 2.4. Western Blotting (WB)

A mixed solution of radio immunoprecipitation assay lysis buffer (RIPA, P0013C, Beyotime, Shanghai, China), phenylmethanesulfonyl fluoride (PMSF, ST506, Beyotime, Shanghai, China), and phosphatase inhibitor (P1081, Beyotime, Shanghai, China) was added to the striatal tissue to extract the total protein. Then, the striatal tissue was thoroughly ground and centrifuged at 13,000 rpm for 5 min at 4 °C, and the centrifuged supernatant was collected and assayed for protein concentration using a BCA protein assay kit (BL521A, Biosharp, Guangzhou, China). The protein solution with a known protein concentration was separated by sodium dodecyl sulfate–polyacrylamide gel electrophoresis, the loading amount was 30 μg of total protein, and then the target protein was transferred to 0.45 μm polyvinylidene difluoride membranes (PVDF, ISEQ00010, Merck, Kenilworth, NJ, USA). The membranes were blocked with 5% (*w*/*v*) skim milk (36120ES76, Yeasen Biotech, Shanghai, China) for 2 h and then incubated with a specific primary antibody (TH, 1:1000, 58844S, CST, Boston, MA, USA) at 4 °C for 12 h. Then, the membrane was incubated with the specific secondary antibody (HRP-conjugated goat anti-rabbit IgG, 1:5000, A0277, Beyotime, Shanghai, China) solution for 2 h. Finally, the band containing the target protein was incubated and reacted with the chemiluminescence detection reagent (P90720, Merck, Kenilworth, NJ, USA), and it was immediately imaged by the chemiluminescence imaging system (Tanon-2500B, Tanon, Shanghai, China). The detection results were quantitatively processed using Image J software (version 1.53k) (NIH, Bethesda, MD, USA).

### 2.5. High-Performance Liquid Chromatography (HPLC)

The levels of neurotransmitters dopamine (DA), 5-hydroxytryptamine (5-HT), and their metabolites in the striatum were determined by HPLC with a fluorescence detector (Waters 2475, Milford, CT, USA). An Atlantis t3 column (150 mm × 4.6 mm, 5 μm, Waters) was used. The mobile phases consisted of ultrapure water, acetonitrile, and 0.01 M phosphate buffer (pH = 4). For sample processing, 0.1 M perchloric acid (10 μL/mg striatum) was added into the striatal tissue. The striatal tissue was then thoroughly ground and centrifuged at 13,000 rpm for 10 min at 4 °C, the centrifugation supernatant was collected and filtered through a 0.22 μm filter, and the filtered sample solution was used for the final assay.

### 2.6. Immunofluorescence (IF) Staining

The fresh brains of the mice were removed, fixed in 4% PFA solution for 24 h, and then dehydrated in 20% and 30% sucrose solutions, each for 24 h. Both tissue fixation and dehydration were performed at 4 °C. Finally, the dehydrated brain tissue was embedded with an optimal cutting temperature compound (O.C.T. Compound, Tissue-Tek, SAKURA, Torrance, CA, USA) at −20 °C. The embedded brains were frozen and sectioned on a cryostat (CM3050S, Leica, Wetzlar, Germany) with the operating temperature set to −20 °C, and the 10 μm-thick brain sections containing the striatum, hippocampus, and SNpc were collected. For IF staining, the sections were first blocked with 10% donkey serum followed by overnight incubation with primary antibody solution containing goat anti-Iba-1 (1:500, ab5076, abcam, Cambridge, UK), mouse anti-GFAP (1:500, MAB360, Merck, Kenilworth, NJ, USA), and rabbit anti-TH (1:500, AB152, Merck, Kenilworth, NJ, USA). Afterward, a secondary antibody solution containing Cy3-conjugated donkey anti-goat IgG (1:1000, A0502, Beyotime, Shanghai, China), DyLight-488-conjugated donkey anti-mouse IgG (1:1000, BA1145, BOSTER, Wuhan, China), and DyLight-488-conjugated donkey anti-rabbit IgG (1:1000, BA1146, BOSTER, Wuhan, China) were used to react at 37 °C for 1 h. The cell nucleuses were then stained using Antifade Mounting Medium with DAPI (P0131, Beyotime, Shanghai, China) [5]. An Axio Imager Z2 fluorescence microscope (Carl Zeiss LSM880, Zeiss, Oberkochen, Germany) was used for the fluorescence imaging. For each mouse, 5 brain sections at specific anatomical locations of the right hemisphere were selected (spaced 100 μm apart) and stained with IF. The number of TH^+^ neurons in the SNpc region (Bregma −2.92 mm to −3.52 mm and spaced 100 μm apart) and the number of GFAP- and Iba-1-positive cells per square millimeter (mm^2^) in the striatum (Bregma 0.14 mm to 1.18 mm and spaced 100 μm apart) and hippocampus (Bregma −1.58 mm to −2.30 mm and spaced 100 μm apart) of the five brain sections from the right hemisphere were then calculated. The average of these five sections was used for the final statistical analysis. Micrographs were processed for quantification using Image J software (version 1.53k) (NIH, Bethesda, MD, USA). Based on the gray value of the micrographs, the threshold was set to fully display the cell morphology. The micrographs were used to quantify the number of GFAP- and Iba-1-positive cells per square millimeter (mm^2^) with the Image J software (version 1.53k).

### 2.7. Enzyme-Linked Immunosorbent Assay (ELISA)

A mixed solution of radio immunoprecipitation assay lysis buffer (RIPA, P0013C, Beyotime, Shanghai, China), phenylmethanesulfonyl fluoride (PMSF, ST506, Beyotime, Shanghai, China), and phosphatase inhibitor (P1081, Beyotime, Shanghai, China) was added to the hippocampal tissue to extract the total protein. Then, the hippocampal tissue was thoroughly ground and centrifuged at 13,000 rpm for 5 min at 4 °C, and the centrifuged supernatant was collected and assayed for the protein concentration using the BCA protein assay kit (BL521A, Biosharp, Guangzhou, China). Each mouse took 10 μL of hippocampal tissue samples for ELISA detection. The concentrations of proinflammatory cytokines in the hippocampus, including TNF-α (FEK0527, BOSTER, Wuhan, China) and IL-1β (EK0394, BOSTER, Wuhan, China), were detected by a commercial ELISA kit. The experimental procedures were carried out in strict accordance with the manufacturer’s instructions. The concentrations of TNF-α and IL-1β were presented as pg/mg protein.

### 2.8. Statistical Analyses

All data are expressed as the mean ± SEM. Statistical differences among the four groups were analyzed using one-way ANOVA with LSD post hoc test by using SPSS 26.0 software. The statistical significance thresholds were set at: * *p* < 0.05, ** *p* < 0.01, and *** *p* < 0.001. All of the quantitative statistical graphs were drawn using GraphPad Prism 8.0 software.

## 3. Results

### 3.1. TMAO Significantly Increased Serum TMAO Levels in the Mice

To explore the effect of high levels of TMAO on PD mice, it was necessary to verify whether the TMAO concentration in the peripheral blood of the mice was successfully increased by the TMAO treatment. Firstly, the concentration of TMAO in the mouse serum was detected by LC–MS. The chemical structural formulas of TMAO and d9-TMAO are shown in Figure 1B,C. Representative ion chromatograms of TMAO and its stable isotope d9-TMAO are shown in Figure 1D. The LC–MS results show that the serum TMAO concentration in the TMAO group and the TMAO + MPTP group was significantly higher than that in the control group and the MPTP group (Figure 1E). These results indicate that TMAO supplied via drinking water significantly increased the serum TMAO levels of the mice.

### 3.2. TMAO Partially Exacerbated Motor Dysfunction in the PD Mice

The motor capacity of the mice in each group was evaluated by the pole test and the traction test. In the pole test, the descent time of the mice in the TMAO + MPTP group was slightly longer than that of the mice in the MPTP group, but there was no statistical difference (Figure 2A), indicating that TMAO had no significant effect on the behavioral retardation of the PD mice. In the traction test, the score of the TMAO + MPTP group was significantly lower than that of the MPTP group, indicating that TMAO impaired the muscle strength and balance force of the PD mice (Figure 2B). Collectively, TMAO exacerbated the motor dysfunction of the PD mice to some extent.

### 3.3. The Effect of TMAO on Dopaminergic Neurons, TH Protein, and Neurotransmitter Levels in the Substantia Nigra and Striatum

Tyrosine hydroxylase (TH) is a key enzyme in dopamine synthesis, and its low expression level is one of the main pathological features of PD [16]. In order to evaluate the effect of TMAO on dopamine neurons and TH expression in the PD mice, we first measured the number of dopaminergic neurons in the SN by IF and the expression levels of the TH protein in the striatum by WB (Figure 3A,C). As expected, MPTP treatment caused a loss of dopamine neurons in the SN and significantly decreased striatal TH protein expression both in the MPTP group and the TMAO + MPTP group (Figure 3B,D), whereas there was no difference in the number of dopaminergic neurons and the striatal TH expression between the TMAO + MPTP group and the MPTP group (Figure 3B,D). Hence, our results suggest that the high serum levels of TMAO did not exacerbate the loss of dopamine neurons nor the reduction in TH protein expression in the PD mice. The levels of DA, 5-HT, and their metabolites in the striatum were further determined by HPLC to explore the effect of TMAO on striatal neurotransmitters. The results show that the MPTP treatment significantly reduced the levels of DA and DOPAC in the striatum (Figure 3E,F). However, there was no statistical difference in the DA levels between the MPTP group and the TMAO + MPTP group (Figure 3E), which was consistent with the results of the WB detection of TH expression in the striatum. It is worth noting that the DOPAC levels in the TMAO + MPTP group were significantly higher than those in the MPTP group (Figure 3F). In addition, the DOPAC/DA ratio and the HVA/DA ratio (indicators reflecting DA metabolism) were significantly higher in the TMAO + MPTP group than in the MPTP group (Figure 3G,H), indicating that TMAO promoted DA metabolism. The neurotransmitter 5-HT is important in the brain in that it can regulate emotion and cognition [17]. For the 5-HT assay, the TMAO + MPTP group showed significantly reduced levels of 5-HT compared with the MPTP group (Figure 3I). Meanwhile, the ratio of 5-HIAA/5-HT significantly increased in the TMAO + MPTP group compared with the MPTP group (Figure 3J). In conclusion, TMAO reduced the levels of the neurotransmitter 5-HT and promoted DA and 5-HT metabolism in the PD mice.

### 3.4. TMAO Increased Activation of Glial Cells in the Striatum of PD Mice

Previous studies have shown that glia-mediated neuroinflammation in the striatum is one of the important pathogeneses of PD [18]. Li et al. showed that TMAO can cross the BBB, activate microglia and astrocytes, and induce neuroinflammation [8,19]. Therefore, in this study, to further reveal the effect of TMAO on neuroinflammation in the PD mice, IF was used to detect the activation of microglia and astrocytes in the striatum of the PD mice. The IF results showed that the Iba-1^+^ microglia and GFAP^+^ astrocytes were significantly activated in the striatum in the MPTP group compared with the control group (Figure 4A–D). More importantly, compared with the MPTP group, Iba-1^+^ microglia and GFAP^+^ astrocytes were activated more significantly in the striatum of the TMAO + MPTP group (Figure 4A–D), indicating that TMAO significantly aggravated the neuroinflammation mediated by the microglia and astrocytes in the striatum of the PD mice. Similarly, for the TMAO treatment alone, the microglia were significantly more activated than those in the control group, but there was no difference in astrocyte activation (Figure 4A–D).

### 3.5. TMAO Increased the Activation of Glial Cells in the DG of the Hippocampus in the PD Mice

The effect of TMAO on hippocampal neuroinflammation has been reported. Herein, in order to explore the effect of TMAO on the hippocampus of the PD mice, Iba-1^+^ and GFAP^+^ glia cells in the hippocampal DG region were detected by IF. As shown in Figure 5, the activated microglia and astrocytes in the hippocampal DG in the MPTP and TMAO + MPTP groups were increased to varying degrees compared with the control and TMAO groups. Specifically, compared with the control group, TMAO significantly induced the activation of astrocytes in hippocampal DG (Figure 5B,D). More importantly, compared with the MPTP group, the microglia and astrocytes in the DG hippocampus were more significantly activated in the TMAO + MPTP group (Figure 5A–D). These results further support TMAO inducing glia-mediated hippocampal neuroinflammation.

### 3.6. TMAO Increased Levels of Proinflammatory Cytokines in the Hippocampus of the PD Mice

We further evaluated the effect of TMAO on hippocampal neuroinflammation in the PD mice by detecting the levels of proinflammatory cytokines (TNF-α and IL-1β) by using ELISA. The results show that, compared with the control group, the hippocampal proinflammatory cytokines were significantly increased in the MPTP group, which is consistent with previous reports [20]. More importantly, the level of proinflammatory cytokines in the hippocampus of the mice in the TMAO + MPTP group was significantly higher than that in the MPTP group (Figure 6A,B). A similar result was observed in the TMAO group, with a significant increase in hippocampal proinflammatory cytokines in the TMAO group compared with the control group. These results suggest that the TMAO treatment induced neuroinflammation in the hippocampus of the TMAO-treated mice, and this proinflammatory effect was more severe and obvious in the TMAO + MPTP group mice, as manifested by higher levels of proinflammatory cytokines than in the MPTP group.

## 4. Discussion

In recent years, more and more evidence has shown that TMAO is closely related to neurodegenerative diseases and contributes to the generation of neuroinflammation [21]. Clinical studies have found abnormal changes in TMAO level in the peripheral blood of PD patients. In 2020, Sankowski et al. reported that the plasma TMAO levels of PD patients were higher than those of the normal population [22]. However, in November of the same year, Chung et al. found that the plasma TMAO levels of PD patients were significantly lower than that of a normal control population [10]. It is worth noting that in February 2020, Chen et al. found that high levels of TMAO in the peripheral blood may be positively correlated with the progression and severity of disease in PD patients [9]. The above clinical studies suggest that there is a significant correlation between TMAO and PD, but the research conclusions are inconsistent. Meanwhile, further experimental studies on the effect of TMAO on PD have not been reported. Based on the above situation, we are here to investigate the effects of high levels of TMAO in the peripheral blood on brain pathology and behavioral disorders in the MPTP-induced PD model mice.

Previous research reported that supplying TMAO in drinking water can successfully increase the blood concentrations of TMAO in mice [23]. In this study, serum TMAO levels were firstly measured by LC–MS. The results showed that we successfully induced a higher level of TMAO in the peripheral blood of the mice. However, the TMAO levels were not significantly higher in the MPTP group compared to the control group. Although higher levels of TMAO in the peripheral blood of the PD patients have been reported, similar phenomena were not observed in the MPTP-induced PD model in this study. This may be attributed to the fact that there are limitations to the animal model itself and species differences. TMAO production in the body is closely related to diet, and in this study, the MPTP group received a standard diet and water that did not directly contain TMAO or its precursors (choline and L-carnitine, which come mainly from red meat and seafood). IF, WB, and HPLC showed that TMAO had no significant effect on the loss of dopamine neurons, TH expression, or DA content reduction in the SN and striatum of the PD mice. Previous studies in our laboratory have shown an increase in the striatum DA metabolic rate in MPTP-induced PD mice [24,25]. In this study, compared with the MPTP group, the DOPAC content, HVA/DA ratio, and DOPAC/DA ratio (DA metabolic rate index) were significantly increased in the TMAO + MPTP group, indicating that TMAO exacerbated DA metabolism in the PD mice. DA is an important neurotransmitter responsible for regulating balance and movement [26]. The degenerated death of nigral dopaminergic neurons in PD leads to a decrease in striatal dopamine levels, which in turn leads to motor dysfunction. The behavioral tests also found that TMAO partially aggravated movement disorders in the PD mice, which may be related to the increase in DA metabolism in the striatum by the TMAO. Meanwhile, under pathological conditions, excessive DA metabolism results in the accumulation of 3, 4-dihydroxyphenacetaldehyde (DOPAL), which has been shown to be a neurotoxic substance [27]. In conclusion, high levels of TMAO in the peripheral blood did not directly aggravate the reduction in dopaminergic neurons, TH protein, or DA but might have promoted the transformation of DA to its metabolites in the brains of the PD mice.

In addition, the proinflammatory effect of TMAO has been reported [28]. For instance, elevated circulating blood TMAO induced by a high choline diet exacerbates microglia and astrocyte activation and neuroinflammation after acute cerebral hemorrhage injury [19]. TMAO is a promoter of inflammation in the brain [29]. Inflammation of the brain, especially neuroinflammation in the striatum, has long been thought to be one of the most important causes of PD [30,31]. The role of microglia- and astroglia-mediated neuroinflammation in PD has been widely studied [32]. In this study, the IF results showed that MPTP can extensively activate microglia and astrocytes in the striatum of PD mice. In particular, the levels of glial cells in striatum activation were more significant in the TMAO + MPTP group than in the MPTP group, suggesting that TMAO exacerbated neuroinflammation in the PD mice. TMAO-induced striatal neuroinflammation may be one of the driving factors for further deterioration of motor ability in PD patients. Meanwhile, the activation of microglia and astrocytes in the SN also preliminarily evaluated by IF assay, and relevant images are presented in Appendix A.

Several studies have revealed that TMAO impairs spatial working memory performance in young mice in behavioral cognition experiments and that TMAO worsened postoperative cognitive decline in aged rats, accompanied by hippocampal neuroinflammation and synaptic structure impairment [11,33]. Similarly, PD is often accompanied by non-motor symptoms such as cognitive decline, dementia, anxiety, and depression, which may be associated with hippocampal impairment [34,35]. Meanwhile, one study reported the activation of microglia and astrocytes in the hippocampus and neuronal damage in MPTP-induced PD models [36]. The DG of the hippocampus is one of the most important regions regulating cognition, emotion, and learning [37]. Therefore, to verify whether the TMAO treatment increased the activation of hippocampus glial cells, we detected Iba-1^+^ and GFAP^+^ glial cells in the DG of the hippocampus by using IF. Consistently, we found that the activation of microglia and astrocytes was significantly increased in the DG of the hippocampi of the TMAO + MPTP group compared with the MPTP group. Activated glial cells usually cause tissue damage by releasing proinflammatory cytokines [38], so we further detected the levels of proinflammatory cytokines in the hippocampus. We showed that the TMAO significantly increased the levels of TNF-α and IL-1β in the hippocampi of both the TMAO group and the TMAO + MPTP group. The above results together show that TMAO may exacerbate neuroinflammation in the hippocampus of PD mice by promoting the activation of glial cells and the release of proinflammatory factors. As a brain region that mainly regulates cognition and memory [39], studies have shown that hippocampal neuroinflammation can impair cognitive function [40]. Thus, further experiments need to be designed to reveal whether high levels of TMAO influence cognition and memory. Studies have shown that TMAO can induce oxidative stress, neuronal aging, and mitochondrial dysfunction in the brain [21]. Brunt, Vienna E et al. also reported that TMAO can directly activate cultured human astrocytes in vitro [11]. In this exploratory experiment, we observed that high serum TMAO increased glial cell response in PD mice. Although the mechanism behind this phenomenon is unclear and requires further investigation, it is reasonable to assume that high serum TMAO crosses the BBB and directly interacts with glial cells, thereby exacerbating neuroinflammation in the PD mice.

## 5. Conclusions

Collectively, this study investigated the potential effects of higher circulating TMAO levels on behavioral disorders, dopaminergic neurons, TH protein expression, neurotransmitter levels, and neuroinflammation in MPTP-induced PD mice. We reported that orally-induced higher levels of TMAO in the peripheral blood aggravated behavioral disorders, reduced the level of neurotransmitter 5-HT, and promoted the metabolism of the neurotransmitters DA and 5-HT in the PD mice. Meanwhile, TMAO significantly activated microglia and astrocytes in the striatum and DG of the hippocampi of the PD mice, as well as increasing the levels of pro-inflammatory cytokines in the hippocampi. In conclusion, the results of this study show that higher circulating TMAO levels exacerbated behavioral disorders, the metabolism of the neurotransmitters DA and 5-HT in the striatum, and neuroinflammation in the striatum and hippocampi of the PD mice. Therefore, higher TMAO levels in the peripheral blood may be one of the risk factors for PD.

## Figures and Tables

**Figure 1 brainsci-13-00790-f001:**
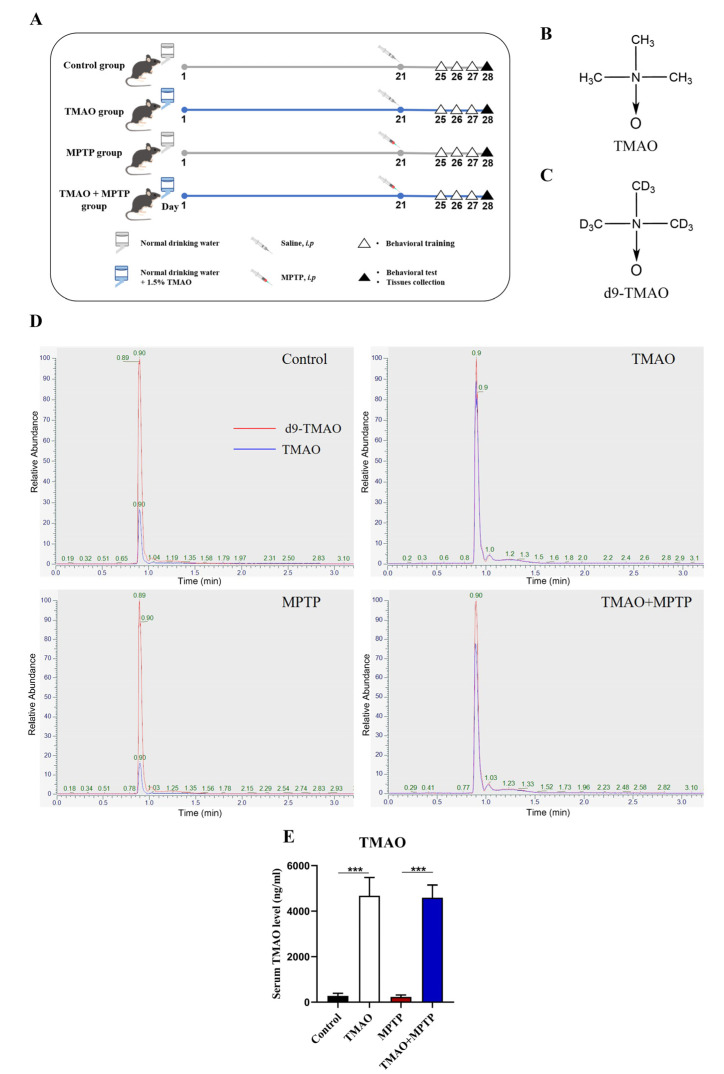
TMAO significantly increased the serum TMAO levels in the mice: (**A**) schematic diagram of the animal experiment; (**B**) chemical formula for TMAO; (**C**) chemical formula for d9-TMAO; (**D**) representative ion chromatograms for LC–MS detection; and (**E**) serum TMAO concentration. n = 5/group. *** *p* < 0.001.

**Figure 2 brainsci-13-00790-f002:**
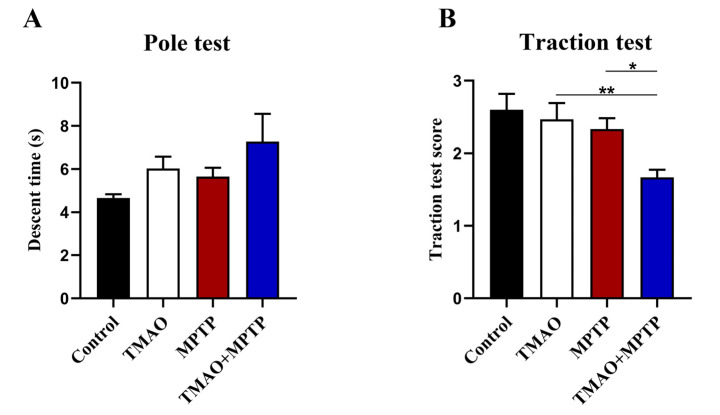
TMAO partially exacerbated motor dysfunction in the PD mice: (**A**) the descent time of mice in the pole test; and (**B**) scores of mice in the traction test. n = 5/group. * *p* < 0.05; ** *p* < 0.01.

**Figure 3 brainsci-13-00790-f003:**
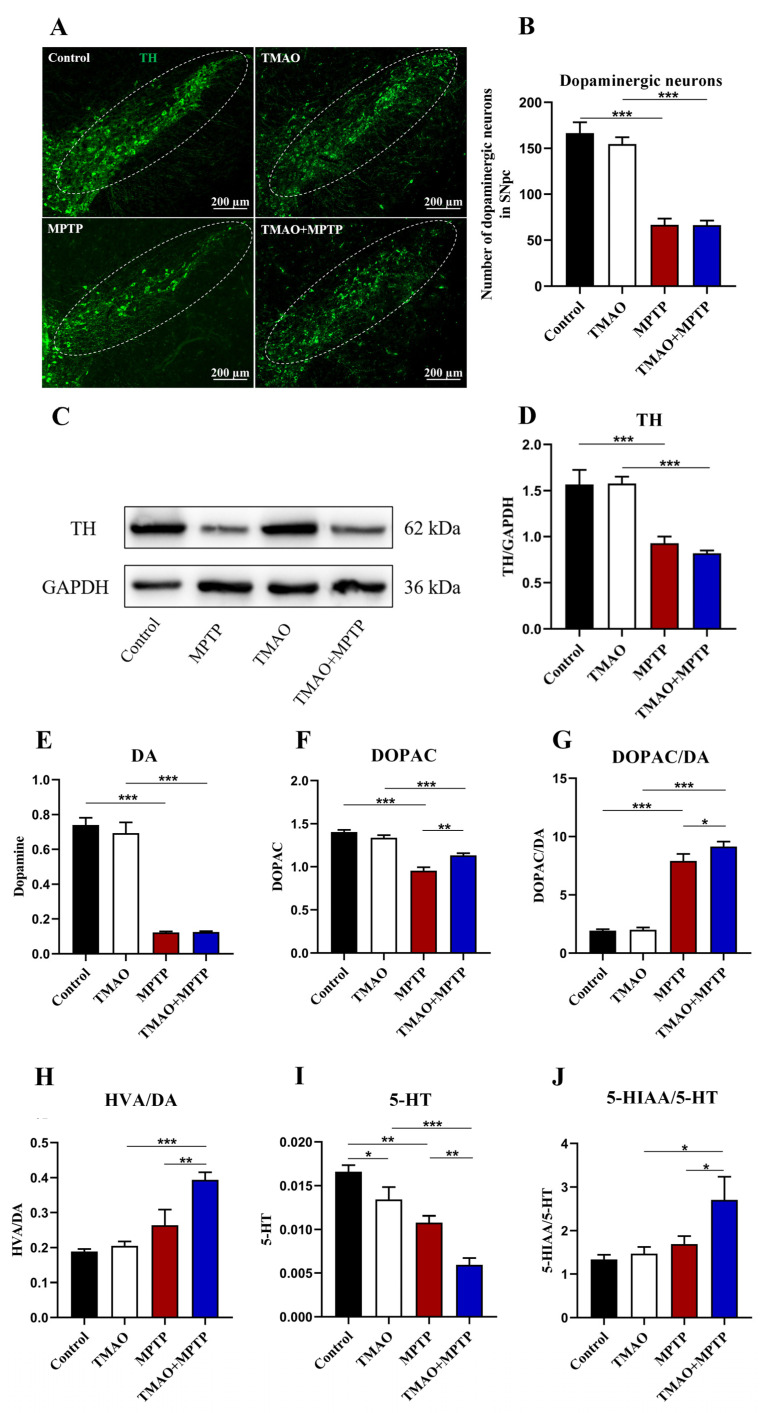
The effect of TMAO on the dopaminergic neurons in the substantia nigra and the striatal TH protein and neurotransmitter levels of the PD mice: (**A**) representative photomicrographs of TH (the dopaminergic neuron marker) staining in the SN (scale bar: 200 μm); (**B**) the counts of TH^+^ neurons in the SN; (**C**) representative images of Western blotting of striatal TH expression; (**D**) the intensity of the bands was quantified with Image J software (version 1.53k) and quantitative data for TH following normalization to GAPDH; (**E**) the concentration of DA in the striatum; (**F**) the concentration of DOPAC in the striatum; (**G**) the DOPAC/DA ratio; (**H**) the HVA/DA ratio; (**I**) the concentration of 5-HT in the striatum; and (**J**) the 5-HIAA/5-HT ratio. For IF: n = 3/group. For WB: TMAO + MPTP group n = 4, and for the other three groups, n = 5/group. For HPLC: n = 5/group. * *p* < 0.05, ** *p* < 0.01, and *** *p* < 0.001.

**Figure 4 brainsci-13-00790-f004:**
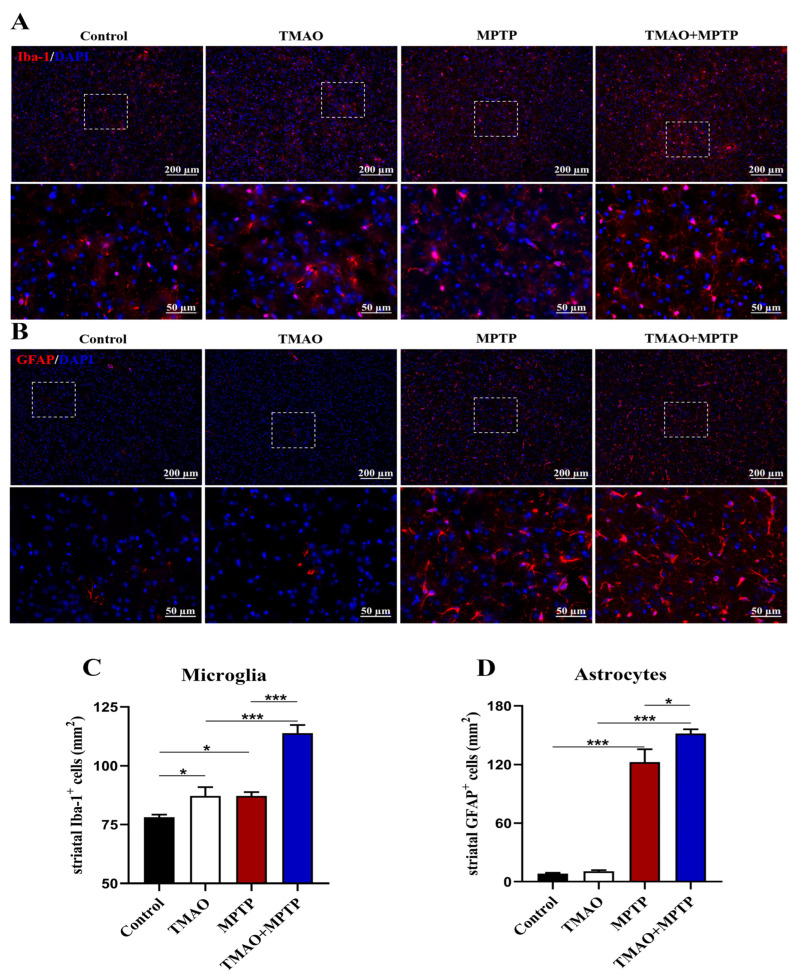
TMAO increased activation of glial cells in the striatum of the PD mice: Representative images of (**A**) Iba-1 (microglial marker, red) and (**B**) GFAP (astroglial marker, red) fluorescent staining in the striatum at 10× magnification. The 40× images corresponding to the white box area are directly below each set of images (scale bar: 50 μm). (**C**) The number of Iba-1^+^ microglia and (**D**) the number of GFAP^+^ astrocytes in the striatum were quantitatively analyzed by Image J software (version 1.53k). The cell nucleuses were stained using antifade mounting medium with DAPI. n = 3/group. * *p* < 0.05 and *** *p* < 0.001.

**Figure 5 brainsci-13-00790-f005:**
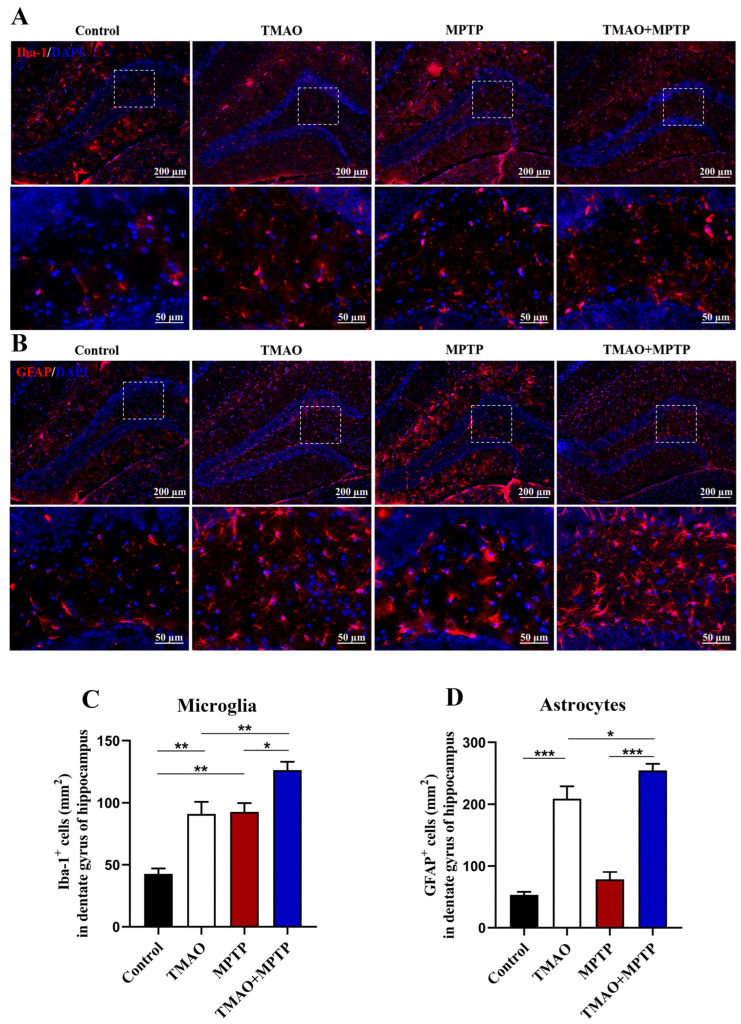
TMAO increased activation of glial cells in the DG of the hippocampus in the PD mice: Representative images of (**A**) Iba-1 (microglial marker, red) and (**B**) GFAP (astroglial marker, red) fluorescence staining in the DG of the hippocampus at 10× magnification (scale bar: 200 μm). The 40× images corresponding to the white box area are directly below each set of images (scale bar: 50 μm). (**C**) The number of Iba-1^+^ microglia and (**D**) the number of GFAP^+^ astrocytes in the DG of the hippocampus were quantitatively analyzed by Image J software (version 1.53k). The cell nucleuses were stained using antifade mounting medium with DAPI. n = 3/group. * *p* < 0.05, ** *p* < 0.01, and *** *p* < 0.001.

**Figure 6 brainsci-13-00790-f006:**
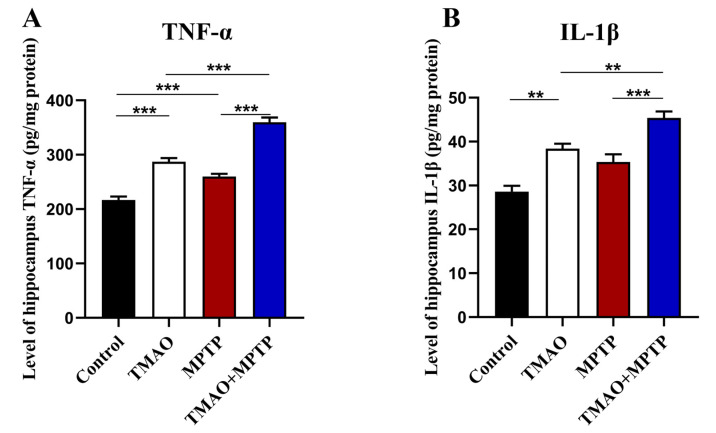
TMAO increased levels of proinflammatory cytokines in the hippocampus of the PD mice: The levels of proinflammatory cytokines (**A**) TNF-α and (**B**) IL-1β in the hippocampus were detected by using ELISA. n = 5/group. ** *p* < 0.01 and *** *p* < 0.001.

## Data Availability

The datasets generated during and/or analysed during the current study are available from the corresponding author on reasonable request.

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
