# Peer review of "Trimethylamine N-Oxide Exacerbates Neuroinflammation and Motor Dysfunction in an Acute MPTP Mice Model of Parkinson’s Disease"

_brainsci, 2023, doi:10.3390/brainsci13050790_

Round 1

Reviewer 1 Report

Comments and Suggestions for Authors

The subject is interesting and has been little explored, however, the manuscript as it is presented requires more work to improve .

The main points to improve are:

1. What was the TMAO dose based on? Did you do a dose-response curve? Did you base it on published literature? Specify in the methods section.

2. Describe behavioral tests further. How is the traction test score determined?

3. No sections describe how the images were processed for the quantification. Why were cells quantified in Iba-1 and GFAP only the MFI? Iba-1+ cell quantification by area? Specify.

4. The ELISA section does not describe how the tissue was processed or the detection limit. Likewise, when dealing with tissue, the data cannot be presented in pg/ml. I assume the hippocampus was homogenized; the proteins in the samples need to be determined (Bradford or Lowry) to present the data pg/mg protein.

5. The objective of the work is to determine the role of TMAO in a Parkinson's model; however, the cytokines were determined in the hippocampus. What is the justification for measuring the cytokines in the hippocampus, not the substantia nigra and striatum, which are the main structures evaluated in the Parkinson's disease models.

6. It is necessary to measure cytokines in the substantia nigra and striatum and not the hippocampus.

7. Figures 4 and 5: the cellular morphological details can not be observed, and the images are of poor quality. Present images with higher magnification that allow seeing the associated morphological changes.

8. It is necessary to quantify Iba-1+ cells per area (mm2)

9. Changes in MFI can be attributed to both morphological changes and increased cell number; please quantify GFAP+ cells per area.

10. Figure 5: Instead of presenting immunofluorescence of the hippocampus, show glial response in the substantia nigra, the main area affected in Parkinson's disease.

11. To be correctly interpreted, cytokine results should be presented as pg/mg of tissue or pg/mg of protein.

12. Authors measured TMAO in serum; can TMAO also be detected in CNS, specifically in the substantia nigra and striatum? TMAO can cause systemic inflammation? Are the effects of TMAO observed in the MPTP model due to the direct impacts of TMAO on glial cells, or are they secondary to peripheral inflammation?

Comments on the Quality of English Language

Quality is sufficient; the text is clear and can be read well.

Reviewer 2 Report

Comments and Suggestions for Authors

In their study, “Timethylamine N-oxide aggravate behavioral disorders and neuroinflammation in PD mice by activating glial cells”, Wei Quan and colleagues investigated the effect of TMAO on MPTP induced Parkinson’s disease (PD) model mice. In particular, the authors describe as the TMAO pretreatment could aggravated the motor deficits in mice and the neuroinflammation response in striatum and hippocampus.

Overall, the study is well designed and deals with a very interesting topic.

The Manuscript is well written and easily intelligible. I have some suggestions for the Authors.

-         In the Immunofluorescence section of Materials&Methods paragraph, authors said: “Micrographs were processed for quantification using Zeiss ZEN 2 (blue edition) software”. However, in the legend of both Figure 4 and 5, authors said: “The number of Iba-1+ microglia and (D) the average fluorescence intensity of GFAP+ astrocytes in the striatum were quantitatively analyzed by Image J software”. Which of the two methods was actually used? It would also be necessary to specify the number of sections analyzed per animal, the corresponding bregma of sections, and the measurements of the analyzed field for each acquired image. Moreover, the method used to calculate the fluorescence intensity of the GFAP signal should also be specified. Why the number of GFAP cells has not been evaluated, as for microglia?

-          In figure 4, the authors should add corresponding higher magnification images to better show cell morphology not appreciable at 10x magnification, as in Figure 5.

-          According to the reviewer should be evaluated the TH+ cells content in the substantia nigra pars compacta (SNpc) and the relative inflammatory response in this brain region by means of GFAP and IBA1 cell content and pro-inflammatory cytokines concentration. The latter parameter should also be considered for the striatum.

-          The authors should  better explain why they analyzed the hippocampus in this PD model.

Minor points:

-          Regarding Figure 1D, the image quality is too low. Please, improve this aspect.

-          Please, add references to DAPI counterstaining (shown in the images of Figures 4 and 5) in the Immunofluorescence section of Materials&Methods paragraph and in the Figures’ Legend (Figures 4 and 5).

-          In the Immunofluorescence section of Materials&Methods paragraph, authors said: “Afterwards, a secondary antibody solution containing Cy3 conjugated donkey anti-goat IgG (1:1000, A0502, Beyotime, China) and DyLight 488 conjugated  donkey anti-mouse IgG (1:1000, BA1145, BOSTER, China) was used to react at 37°C for 1  hour”. Why do both IBA1 and GFAP markers show a red signal in the images?

Reviewer 3 Report

Comments and Suggestions for Authors

In this manuscript, Quan et al., evaluated the behavioral and neuroinflammatory effect of TMAO on MPTP induced-PD mice. This is an interesting manuscript, which is overall very well written and documented.

Major Comments

Since the level of TMAO in PD patients is not well defined (reference 10), also the results showed a decreased level of TMAO in MPTP group compared to other groups. How do authors explain this scenario? And why do you treat mice with TMAO orally? I think, if you see changes in MPTP group, then it is more relevant to this field.

Have you determined the population of microbiota or LPS after MPTP administration.

What is the reason for high TMAO in PD patients? Is this due to the permeability changes of the intestine or increased microbiota population in PD patients. The results showed significant changes in most of the parameters in MPTP+TMAO group compared to other groups, even MPTP alone or TMAO. Authors need to explain in the manuscript what is the mechanism of combined effect of MPTP+TMAO in gut microbiota or neuroinflammation and why TMAO level was not increased in MPTP group.

Reviewer 4 Report

Comments and Suggestions for Authors

Title: The title is not appropriate and needs modification. It could be Trimethylamine N-oxide exacerbates neuroinflammation and motor dysfunction in an acute MPTP mice model of Parkinson’s disease

Abstract: Needs modification. Background of Parkinson’s disease and clinical relevance to clinical gut brain axis is missing

Introduction: Modification suggested. Acute MPTP model an inflammatory model of PD may be described in brief.

Materials and Methods: Did the authors check the level of microglial activation/ inflammation after 48 hours of acute MPTP intoxication? Also, not sure why the substantia nigra sections were not included. Immunohistochemistry and quantification of glial activation is recommended. Tyrosine hydroxylase immunohistochemistry of substantia nigra neurons and quantification could have added more insight to the role of trimethylamine N-oxide in PD.

Results:

The clarity of Figure 1D could be increased.

Discussion:

Agreed that studies have shown that hippocampal neuroinflammation can impair cognitive function [32]. More than the involvement of hippocampus, the direct role of substantia nigra and striatum in motor dysfunction could have been discussed in detail.

Comments on the Quality of English Language

Language:  Major grammatical corrections must be made throughout the manuscript. There are several typos and spelling errors including the title.

Major language editing is needed in the revised paper.

Round 2

Reviewer 1 Report

Comments and Suggestions for Authors

The authors attended my observations as time allowed it.

I still believe that cytokines should be measured in the substantia nigra and striatum to integrate the results in this model. I understand that time is a critical limitation.

Images improved, and authors showed preliminary immunofluorescences regarding Iba1 and GFAP in the substantia nigra. Consider including these images as supplementary material or finish de n to be able to quantify cells.

Some minor details:

line 13: write trimethylamine

line 66: include a MPTP

line 68: exhibits

line 69: include which is one.....

line 69: missing reference

line 73: use induce instead of construct

line 158: add a comma after hippocampus

line 161: add a comma before and

line 165: add a comma before and

line 166: were stained

line 264: correct striatal

Comments on the Quality of English Language

Check the new paragraphs to find possible mistyping.

Reviewer 2 Report

Comments and Suggestions for Authors

I have no new comments for the authors

Reviewer 3 Report

Comments and Suggestions for Authors

The manuscript is substantially revised 

Reviewer 4 Report

Comments and Suggestions for Authors

Manuscript is substantially revised 

Comments on the Quality of English Language

Minor language correction is needed. 
